# SNOW: Subscribing to Knowledge via Channel Pooling for Transfer & Lifelong Learning of Convolutional Neural Networks

**Chungkuk Yoo**[1]    **Bumsoo Kang**[1,2]    **Minsik Cho**[1]
[1]IBM, Austin TX, USA    [2]KAIST, Daejeon, South Korea
ckyoo@ibm.com    bumsoo@nclab.kaist.ac.kr    minsikcho@us.ibm.com

## Abstract

SNOW is an efficient learning method to improve training/serving throughput as well as accuracy for transfer and lifelong learning of convolutional neural networks based on knowledge subscription. SNOW selects the top-K useful intermediate feature maps for a target task from a pre-trained and frozen source model through a novel channel pooling scheme, and utilizes them in the task-specific delta model. The source model is responsible for generating a large number of generic feature maps. Meanwhile, the delta model selectively subscribes to those feature maps and fuses them with its local ones to deliver high accuracy for the target task. Since a source model takes part in both training and serving of all target tasks in an inference-only mode, one source model can serve multiple delta models, enabling significant computation sharing. The sizes of such delta models are fractional of the source model, thus SNOW also provides model-size efficiency. Our experimental results show that SNOW offers a superior balance between accuracy and training/inference speed for various image classification tasks to the existing transfer and lifelong learning practices.

## 1 Introduction

Learning new tasks from old tasks over time as natural intelligence does is a key challenge in artificial intelligence, and transfer and lifelong learning are two popular strategies in this direction. Transfer learning delivers a neural network with good predictive power by duplicating and tuning the parameters for a pre-trained source model against a dataset for a new task (Dauphin et al., 2012; Donahue et al., 2014). However, transfer learning from a source task to many target tasks incurs overall significant training and space overhead due to multiple large models to customize and store (Mudrakarta et al., 2019). On the other hand, lifelong learning can enable substantial parameter sharing and deliver multiple target tasks with less training time and smaller model size, but may suffer from catastrophic forgetting or lower accuracy (McCloskey & Cohen, 1989). Comprehensive efforts to tackle such challenges have been proposed but at significant computational overhead in general (Rusu et al., 2016; Guo et al., 2019; Mudrakarta et al., 2019; Li & Hoiem, 2018).

In this work, we propose SNOW for efficient transfer and lifelong learning, which consists of an inference-only source model, multiple task-specific delta models, and channel pooling in-between. Unlike transfer learning, we let multiple delta models subscribe to a source model through a channel pooling layer. SNOW is fundamentally different from the prior arts in transfer and lifelong learning in the following aspects: **a)** the source model is frozen during both training and inference, **b)** the top-K intermediate feature maps of the source model are learned for each delta model using channel pooling powered by Gaussian reparametrization technique (Kingma & Welling, 2014). And, SNOW does require neither persistent training data nor episodic memories (Sprechmann et al., 2018; Li & Hoiem, 2018) to overcome catastrophic forgetting. Table 1 compares SNOW with prior arts in transfer/lifelong learning, and Fig. 1 visualizes the key structural differences between SNOW and transfer/lifelong learning where the source model for task0 is leveraged to build the target models for task1-3 without persistent datasets as follows:

| | FO[a] | FE[b] | FT[c] | MP[d] | LF[e] | PN[f] | SNOW |
|---|---|---|---|---|---|---|---|
| Overall Accuracy | poor | medium | good | good | medium | good | good |
| Catastrophic forgetting | no | no | no | no | yes | no | no |
| Training efficiency | good | good | poor | poor | poor | poor | good |
| Serving efficiency | good | good | poor | poor | good | poor | good |
| Model-size efficiency | good | good | poor | good | good | poor | good |

[a] Final output only: re-training the final output layer of the duplicated source model (Yosinski et al., 2014).
[b] Feature extraction: re-training a few output layers of the duplicated source model (Whatmough et al., 2019).
[c] Fine-tuning: re-training the entire duplicated source model (Dauphin et al., 2012).
[d] Model patch: training only the patch layers inserted into the source model (Mudrakarta et al., 2019).
[e] Learning w/o forgetting: regularizing hidden activations without persistent datasets (Li & Hoiem, 2018).
[f] Progressive Net: propagating combined feature maps via lateral connections (Rusu et al., 2016).

Table 1: Comparison of various transfer/lifelong schemes and SNOW.

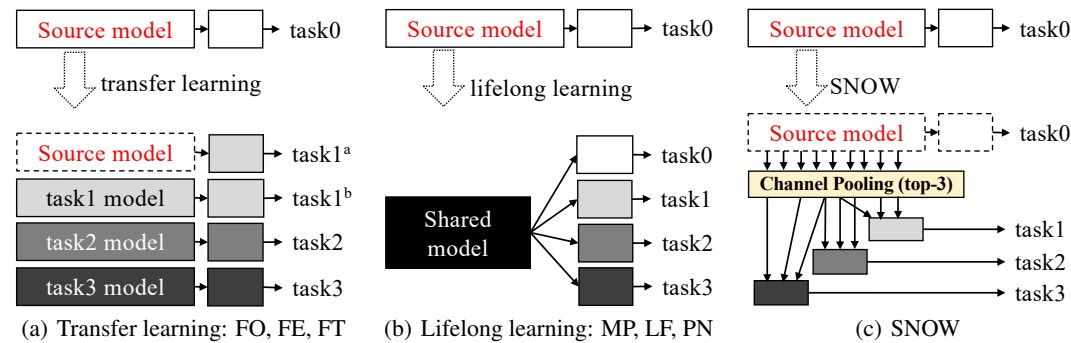

(a) Transfer learning: FO, FE, FT    (b) Lifelong learning: MP, LF, PN    (c) SNOW

Figure 1: SNOW vs. transfer/lifelong learning to build task1-3 from task0. The dotted box indicates a model which stays frozen (or no parameter update), skipping back-propagation during training.

- Transfer learning in (a) tunes the parameters in a copy of the source model for a new task. In transfer learning, updating more parameters improves the prediction accuracy but at higher computational overhead (Yosinski et al., 2014). For example, task1[a] can be obtained faster (as the source model is frozen), but may show lower accuracy than task1[b] where the entire model is customized. Therefore, when high accuracy desired, transfer learning can involve significant training and storage overhead as separate models are trained for different target tasks.

- Lifelong learning in (b) allows a common structure for all the tasks, increasing the parameter efficiency of the entire system. Although lifelong learning could be a faster methodology than conventional transfer learning for multi-task delivery, it may still demand quite some training time as the shared part needs to get trained for all the target tasks and/or can suffer from catastrophic forgetting (unless expensive and complex persistent dataset or episodic memory is available). In (Rusu et al., 2016), a constructive method for lifelong learning is proposed to avoid catastrophic forgetting but at substantial computational and memory footprint overhead.

- As in (c) for SNOW, the source model serves as a static feature supplier (i.e., no back-propagation) to all the target-specific models or delta models (significantly enhancing training and serving speed). Hence, each delta model generates only a handful number of its own local features and obtain all other features for free (greatly improving parameter efficiency). However, instead of supplying all the features to the delta models, SNOW has a channel pooling layer which learns and selects the top-K features for each individual delta model (delivering excellent prediction). For an example of (c), the channel pooling layer independently selects the top-3 intermediate feature maps for task1-3. Unlike conventional lifelong learning, SNOW does not suffer from catastrophic forgetting, as each delta model is independent of each other.

- Cost benefits on SNOW architecture are threefolds. First, training time and memory footprint can be significantly reduced since the source model runs only in an inference mode (thus no back-propagation). Second, the overhead of channel pooling is negligible because it requires only one parameter for each feature map of the source model. Lastly, a delta model needs to be

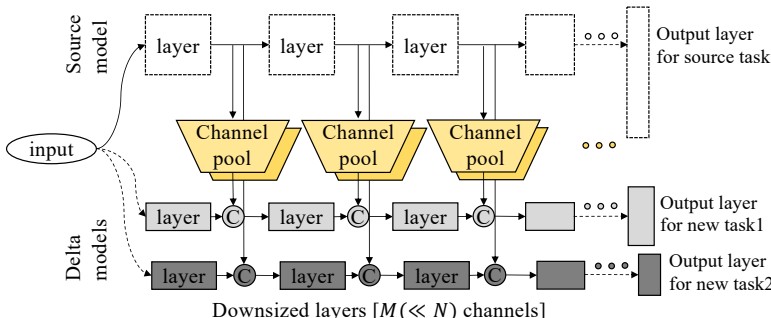

Figure 2: In SNOW architecture, the source model produces intermediate features and the delta models selectively subscribe to them via channel pooling in their training and serving processes. The local features from a delta model and the subscribed features are combined through concatenation.

large enough to produce target-specific features, but can be much smaller than the source model. Fusing those features with the $K$ features from the source model leads to significant parameter count saving compared with existing transfer learning practices.

The rest of the paper is organized as follows. Section 2 details SNOW, and experimental results are in Section 3. Review on prior arts is in Section 4, followed by the conclusion in Section 5.

## 2 SNOW ARCHITECTURE

In this section, we present SNOW which aims scalable training/serving and high accuracy without catastrophic forgetting, as an efficient framework for transfer and lifelong learning. We will describe SNOW from two perspectives: **a)** how to connect the source model and the delta models efficiently (discussed in Section 2.1) and **b)** how to effectively subscribe to the intermediate features from the source model from the delta models for higher accuracy (discussed in Section 2.2). The above two challenges look orthogonal, yet highly correlated, as over-subscription of features degrades model-size efficiency while under-subscription would harm the model accuracy.

### 2.1 ARCHITECTURE AND LATERAL CONNECTIONS

SNOW proposes an architecture where a pre-trained/frozen source model and independently to-be-trained delta models for target tasks are connected via channel pooling which enables uni-directional knowledge flow or knowledge subscription from the source to the delta models as in Fig. 2. The goal of uni-directional knowledge subscription is to capture universal features at one-time computation cost which would be amortized across multiple target tasks, assuming the source model has been trained with a large and comprehensive dataset (as in a typical transfer learning scenario). Meanwhile, target task-specific features are computed in the delta models with local feature extractors. In SNOW, therefore, the delta neural network can be at a much smaller size than the source model as long as the source model can provide a comprehensive set of feature maps. Since these two kinds of features are fused and further co-processed in the delta models, we can intuitively see that the delta models are encouraged to learn a set of task-specific features complementary to the subscribed features. In case a delta model cannot extract a sufficient amount of features from a source net, the delta model capacity would need to grow in order to generate the target-specific features sufficiently by itself. However, making the delta model too large can degrade parameter efficiency and further result in parameter o over-fitting when the target dataset is small, like any other neural network (Heaton, 2008).

In detail, the source model produces intermediate features for a given sample during its feed-forward path (which is the same as inference). The delta model then is trained to use some of these feature maps as inputs for the next layer. The input for $L_i$ layer of the delta model is given as the concatenation of the subscribed features from the source model and the output features from $L_{i-1}$ layer of the delta model. In our experiments (see Section 3 for details), we used a down-sized version of the source model for the delta models, but there is no restriction on how to design a delta model because concatenation is more flexible than feature superposition. More specifically, we used ResNet50 (He

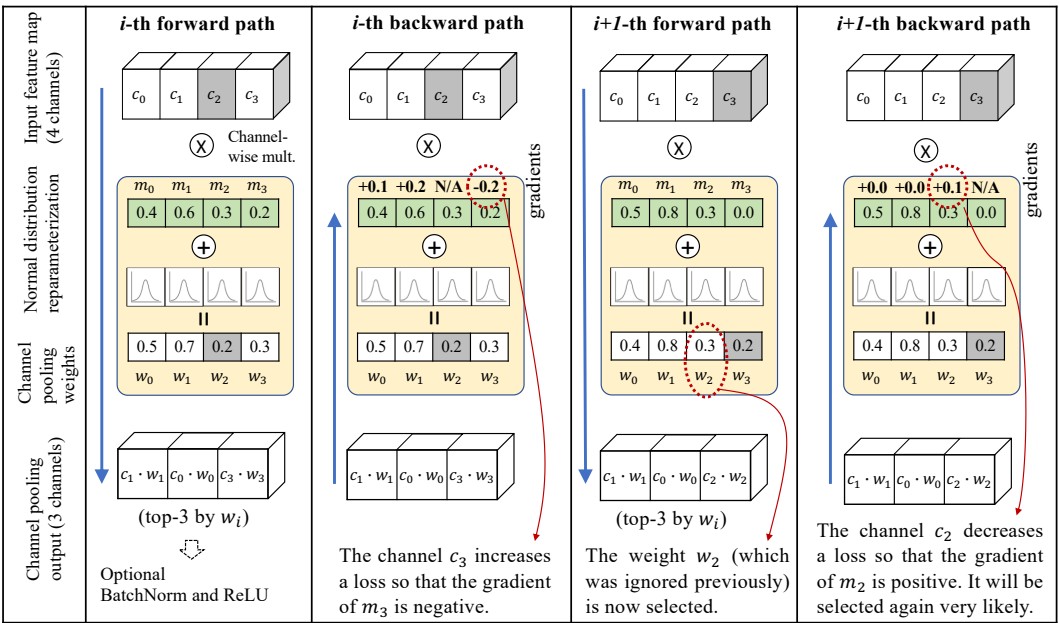

Figure 3: Forward and backward propagation paths for channel pooling.

et al., 2016) trained by ImageNet (Deng et al., 2009) as a source model, and used the same ResNet50 architecture as a delta model for a new task after reducing its channel size by a factor of eight. Note that such lateral connection has been proposed in existing works (Feichtenhofer et al., 2018), but the context is vastly different in a sense that it is mainly to accelerate the inference of a single task during a neural network architecture design, while SNOW is for efficient transfer and lifelong learning.

In our training process for SNOW, all parameters in the source model are frozen while the parameters in the delta models are updated from random initialization. Therefore, we do not need to compute back-propagation for the source model parameters, greatly improving training memory footprint (i.e., nothing to keep for gradient computation) and throughput. And, the delta models have much fewer parameters than the source model. As a result, SNOW training offers much shorter training time and demands less computing resources compared with other conventional transfer/lifelong learning techniques, e.g., FT, MP, LF, PN (see Table 1). Furthermore, we also minimize inference time for multiple tasks by sharing the intermediate features of a single source model across all the target tasks.

The serving efficiency of SNOW is maximized when all the target tasks take the same input. By sharing a feature map from a source network across more tasks, one-time feature map generation overhead can be amortized better, increasing the overall computational efficiency of SNOW architecture. Such single-input-multi-output scenario is highly realistic in practice: for instance, multiple neural networks take in the feed from a single camera, microphone, and sensors (Mathur et al., 2017).

## 2.2 SELECTIVE FEATURE SUBSCRIPTION BY CHANNEL POOLING

The purpose of channel pooling is to find a small subset of intermediate features from a source model in a way that the prediction accuracy from the delta model can be maximized. Therefore, each delta model will be accompanied by one channel pooling layer. In order to find the best subset, each channel pooling layer has a learnable scalar for each source feature map, and SNOW selects the top-K channels to limit the computation and model-size overhead in a stochastic way using normal distribution reparameterization. During inference, SNOW directly feeds the best K feature maps based on the training result without a stochastic step.

Fig. 3 shows the forward and backward paths inside a channel pooling layer. At $i$-th forward path, we first compute the following for any channel $x$: $w_x = m_x + \mathcal{N}(0, \sigma^2)$. Then, we select the top-3 weights $(w_0, w_1, w_3)$ in white and perform channel-wise multiplication between the selected weights and corresponding channels at the same indices so that the results can be consumed by the

corresponding delta model. The purpose of stochasticity is to provide every channel with enough chance to be selected, but we keep $\sigma$ small that the channels can be easily separable. For example, even if $m_2 > m_3$, $w_2$ is ignored at this iteration due to stochasticity.

In the $i$-th backward path, the gradients for all the mean variables are computed **except** $m_2$, as $w_2$ did not participate in the forward execution. Thus, the computed gradients will update $(m_0, m_1, m_3)$ accordingly, and affect the top-K selection in the $(i + 1)$-th forward path. Assume that the gradient for $m_3$ is -0.2, as the $c_3$ hurts a specific target task, then $m_3$ becomes 0.0 (now the lowest).

In the subsequent $i + 1$-th forward path, $(w_0, w_1, w_2, w_3)$ are computed in the same way. This time, the difference between $m_3$ and $m_2$ is big enough to keep $w_3$ from being selected. If $c_2$ is indeed helpful to the target task, then $m_2$ will get bigger, which would in turn help $w_2$ be selected again in the subsequent iterations for further tuning. In this way, while the less relevant feature maps (to a target task) are to be ignored with high probability over time due to the decreased channel pooling weights, the critical channels will not only survive the top-K filtering but also be thoroughly tuned to minimize the training loss. Note that we remove the stochastic part, $\mathcal{N}(0, \sigma^2)$ after training is over so that we directly select top-K trained weights for an inference task.

Determining $K$ (i.e., the number of feature maps to subscribe) has a critical impact on both size and prediction accuracy of the target models. If a delta model undersubscribes to the intermediate features, it would not have sufficient information for higher prediction power. If a delta model oversubscribes, it would require more neurons and computational resources to process larger feature maps, not to mention that it could introduce unwanted noises to the delta model.

|  | $1 \times 1$ conv | Channel-wise attention | **Channel pooling** |
|---|---|---|---|
| Model-size overhead | $O(N \times K)$ | $O(N^2)$ | $O(N)$ |
| Computation pattern | convolution | matrix mult. | ch.-wise scalar mult. |

Table 2: Overhead of each channel subscription approach for a feature map of $N \times h \times w$ size.

There are naive alternative approaches to selectively subscribe to features such as $1 \times 1$ convolution (Rusu et al., 2016) or channel-wise attention (Woo et al., 2018) as in Table 2, but they require more parameters/computations and/or offer worse accuracy than the proposed channel pooling (see Section 3.4 for comparison). In detail, when downsizing a feature map of $N \times h \times w$ size to $K \times h \times w$, a channel pooling layer needs $N$ parameters with $K \times h \times w$ multiplications, while $1 \times 1$ convolution requires $N \times K$ parameters with $N \times K \times h \times w$ multiplications. The channel pooling operation may look similar to the channel-wise attention (Woo et al., 2018), but its attention block keeps the channel size the same as the number of given feature maps, while the channel pooling layer prunes irrelevant features out to have a smaller delta model. Finally, the size of the channel attention module is $O(N^2)$ which is bigger than that of channel pooling $O(N)$, and channel-wise attention requires expensive matrix multiplication, but channel pooling only needs channel-wise scalar multiplication.

## 3 EXPERIMENTS

We implemented **SNOW** in Pytorch 1.1 (Paszke et al., 2017), and compared its training and serving performance with prior arts in terms of accuracy and computational overhead in the following setup;

- Source model: a pre-trained ResNet50 with ImageNet available in `torchvision`.

- Workload: five classification tasks for Food-101 (Bossard et al., 2014), Describable texture dataset (DTD) (Cimpoi et al., 2014), Stanford 40 actions (Yao et al., 2011), Car dataset (Krause et al., 2013), and Caltech-UCSD birds-200-2011 (CUB) (Wah et al., 2011) (see Table 3 for details) without any persistent dataset storage.

- Comparison: **SNOW** and five other transfer/lifelong schemes **FO, FE, FT, MP, PN** in Table 1 (implemented per the corresponding reference in the table note) for training the models. We

| Dataset(#class) | Food(101) | DTD(47) | Action(40) | Car(196) | CUB(200) |
|---|---|---|---|---|---|
| train/test | 75,750/25,250 | 3,760/1,880 | 4,000/5,532 | 8,144/8,041 | 5,994/5,794 |

Table 3: Datasets for experiments.

could not compare with **LF** because **LF** exposed catastrophic forgetting on the tested datasets, preventing fair comparison (see Appendix for more discussion and LF results).

- Environment: IBM POWER9 @ 3.10GHz (44 cores) machine with Red Hat Enterprise Linux Server 7.6, CUDA10.1, CUDNN7.5 and 4 Tesla V100 GPUs (16GB on each).

### 3.1 TRAINING PERFORMANCE

In this section, we report the training performance of all the experimented schemes in terms of training throughput, GPU runtime memory footprint, and Top-1 test accuracy. We trained each algorithm on a single GPU but with a different batch size in order to fit each algorithm's memory footprint into a single GPU memory capacity. While we set the initial learning rate as 0.1 for all other schemes under comparison, we applied 1.0 to **SNOW**, as **SNOW** has channel pooling parts that need training from scratch. We decayed the initial learning rate by a factor of 0.1 per every 30 epochs. For the exact batch sizes and initial learning rates, please refer to Figure 4. We used a SGD optimizer with a momentum of 0.9 and a weight decay factor of $10^{-4}$. We set $K = N/8$ for $N$ intermediate channels from ResNet50 and $\sigma^2 = 0.001$, except for the Action dataset (Yao et al., 2011) ($\sigma^2 = 10^{-8}$). For each algorithm in Table 1, training is done in a sequential manner for all the datasets to reflect the nature of transfer and lifelong learning.

**SNOW** shows the best balance between the training throughput and prediction accuracy over all the tested datasets as shown in Figure 4. Overall, **SNOW-256** delivered a comparable accuracy with up to 2.4x higher throughput than **FT** or **MP** and 5.2x more than **PN**. This implies that SNOW enables transfer or lifelong learning to be completed significantly faster under the same computing resources, yet preserving its quality.

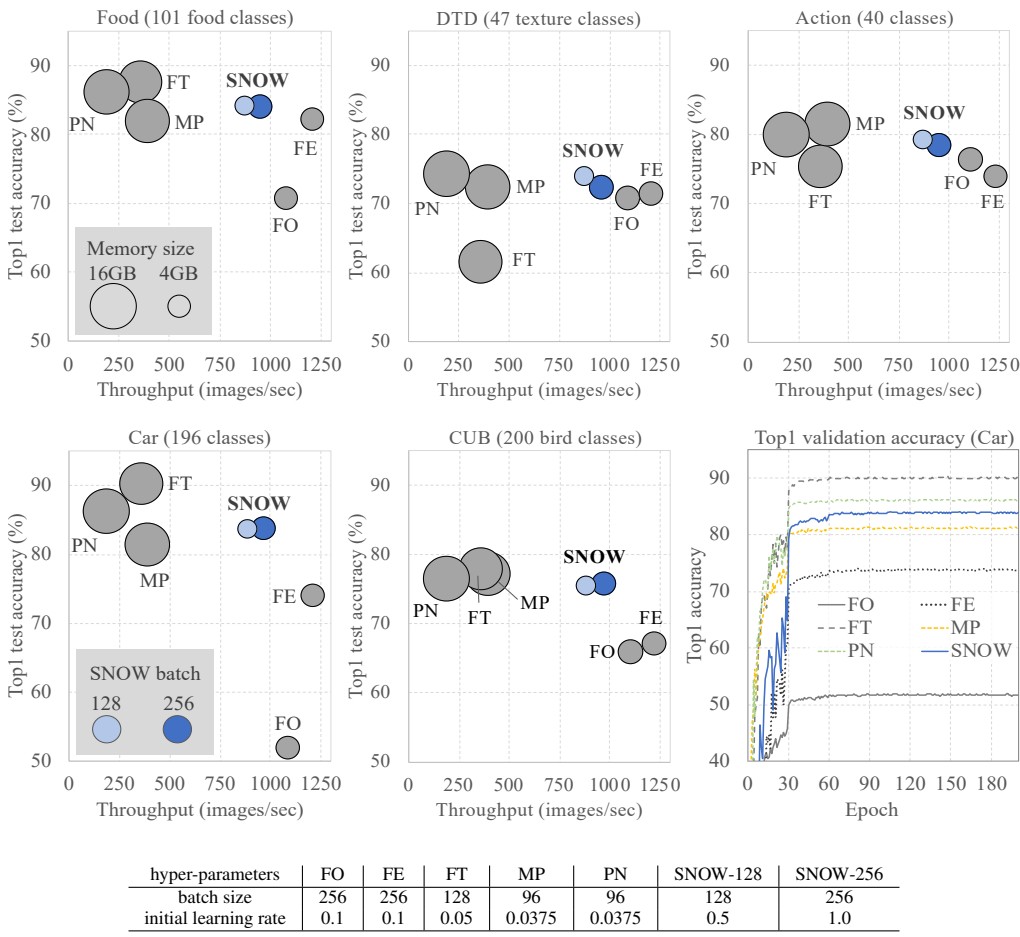

| hyper-parameters | FO | FE | FT | MP | PN | SNOW-128 | SNOW-256 |
|---|---|---|---|---|---|---|---|
| batch size | 256 | 256 | 128 | 96 | 96 | 128 | 256 |
| initial learning rate | 0.1 | 0.1 | 0.05 | 0.0375 | 0.0375 | 0.5 | 1.0 |

Figure 4: Training performance comparison on five different datasets on a single GPU.

The reason why **FT** and **MP** show poor throughputs compared with other light-weight approaches is because nearly full back-propagation is required for both cases. Especially, **MP** is slower than **FT** (despite that majority of parameters are frozen) because it has more layers to deal with yet back-propagation needs to reach out the very first patched layer in the model (thus, still need to compute the gradients for the intermediate features along the way). Meanwhile, **SNOW** performs back-propagation only for the delta models which are substantially smaller than the source model, resulting in significant reduction in training time and computing overhead, e.g., memory footprint for a given batch size. (note that the circle size in Figure 4 represents the memory footprint size). Although **PN** does not require back-propagation on the source model either, its large target model size and heavy lateral connections make it the slowest scheme (Chaudhry et al., 2019).

**FO** and **FE** show marginally higher throughput than **SNOW-256**, but we noticed significant accuracy drops up to 31.8% with **FO** and 9.7% with **FE** from **SNOW-256**'s accuracy. Both approaches look similar to **SNOW** in a way of attaching a small size model and training it while the source model is frozen. However, instead of subscribing to intermediate features, they use only a single feature map of the third residual block (**FE**) or the last residual block (**FO**) of the source model, which prevents them from exploiting the intermediate features. From this comparison, we can see that feature subscription actually contributes to higher accuracy for a new task as we expected in Section 2, yet at a small throughput overhead with cost-efficient channel pooling.

Figure 4 (bottom right) shows that **SNOW** as well as all other techniques converge around 60 epochs. Since the end-to-end training time is is proportional to the number of epochs to run and inversely proportional to the throughput, **SNOW** is much faster than **FT**, **MP**, and **PN** in terms of time-to-convergence, because of 2x~5x higher throughput for the same number of epochs.

## 3.2 Hyper-parameter Sensitivity

In this section, we discussed the hyper-parameter sensitivity of **SNOW** in terms of model accuracy with the Car dataset. We performed additional training on the Car dataset while varying the $K$, the delta model size, and $\sigma^2$, one at a time to understand their impacts on the accuracy. We summarized the outcomes in Table 4 (where $M$ is the source model size) and made the following observations with practical strategies for hyper-parameter tuning:

| Parameter changes | $K$ | | | delta model size | | | $\sigma^2$ | | | |
|---|---|---|---|---|---|---|---|---|---|---|
| | $\frac{N}{4}$ | $\frac{N}{8}$ | $\frac{N}{16}$ | $\frac{M}{4}$ | $\frac{M}{8}$ | $\frac{M}{16}$ | $1e^{-1}$ | $1e^{-3}$ | $1e^{-5}$ | $1e^{-7}$ |
| Top-1 accuracy | 83.10 | 83.79 | 83.39 | 79.02 | 83.79 | 80.36 | 81.27 | 83.79 | 83.45 | 83.35 |
| Top-5 accuracy | 96.93 | 96.94 | 96.93 | 96.00 | 96.94 | 95.93 | 95.85 | 96.94 | 96.99 | 96.88 |

Table 4: Accuracy changes over varying hyper-parameter settings.

- The accuracy is a bit sensitive to the $K$ values. As discussed in Section 2.2, both under-subscription and over-subscription can degrade predictive power: under-subscription means insufficient feature sets for a delta model, while over-subscription may introduce unwanted noises to a delta model. Since over-subscription also incur model-size overhead, it would be desirable to start with small $K$ values first until the peak accuracy gets reached.

- It shows that the performance is sensitive to the delta model size, and clearly exposes the existence of the ideal size. It is obvious that the same rule of thumb in neural network architecture design applies here too: having too few target-specific features hurts accuracy, but having too many (or too big delta net) may lead to over-fitting.

- Too large or too small $\sigma^2$ can lead to sub-optimal predictive power. Therefore, it is important to develop a method to find good sigma values. We're currently researching on the idea to examine the early weight distribution (i.e., for a few epochs) and then determine the sigma value to balance out exploration and stabilization.

## 3.3 Serving Performance

We measured the computation costs of an inference process by serving all six tasks concurrently (i.e., classifiers for ImageNet and the other five in Table 3) against the same input (Zamir et al., 2018;

| Serving approach | ImageNet, Food, DTD, Action, Cars, CUB | | | | | |
|---|---|---|---|---|---|---|
| | FO | FE | FT | MP | PN | **SNOW** |
| Total model parameters (M) | 26.8 | 28.3 | 144.3 | 26.9 | 257.4 | 30.8 |
| Throughput (images/sec) | 7060.8 | 5626.6 | 632.9 | 304.3 | 169.5 | 4731.0 |

Table 5: Computation costs for inference for the ImageNet (source model), Food, DTD, Action, Car, and CUB. We run all tasks together on a single Tesla V100.

Mathur et al., 2017). Overall inference results are summarized in Table 5 and indicate that **SNOW** has significantly less overhead than other high-accuracy approaches such as **FT**, **MP**, **PN** in terms of total model size and inference throughput.

- **FO** is the smallest and shows the best inference throughput due to its architectural simplicity, but please note that **FO** has the worst accuracy on all five datasets.
- **SNOW** yields comparable inference throughput with **FE** (yet shows better accuracy as in Fig. 4) with slightly more parameters.
- **FT** has the 2nd biggest penalty in terms of the parameter count as expected, because each task needs a full set of parameters tuned for a specific task.
- **MP** does not have much overhead in terms of the total model size, but suffers from the poor inference performance. Such penalty comes from the expensive scatter-gather computation patterns which are required to re-use the source model parameters, in addition to the increased number of layers on the critical execution path.
- **PN** has the biggest penalty in terms of the total number of parameters as expected size which leads to poor inference throughput.

In short, we can observe that **SNOW** has critical advantages in a multi-task setup, as it marginally increases the total model size by sharing the source model among all the individual tasks and keeps the inference performance intact, not to mention that it delivers high-quality accuracy.

### 3.4 CHANNEL POOLING EFFECT

In this section, we first examine how the top-K selection changes during the course of training, and thencompares the channel pooling scheme in **SNOW** with other popular dimensional reduction and feature amplification methods in deep learning.

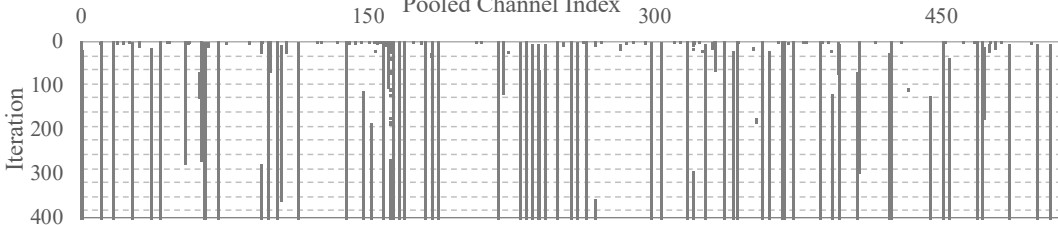

Figure 5: Selected channel index distribution for the Car dataset during training.

Figure 5 shows how the top-K features selected by the proposed channel pooling for the Car dataset change throughout training where the X-axis is for the channel indices, the Y-axis is for the training progress (in terms of iteration), the solid vertical lines represent the selection of the corresponding channels, and the horizontal dotted lines indicate the start of a next epoch. Overall, we found that the top-K feature selections change frequently in the beginning of the training for exploration but get stabilized after additional epochs, as discussed in Section 2.2.

We applied such alternatives as feature subscription in **SNOW** and compared performances in terms of parameter overhead and test accuracy as summarized in Table 6 which clearly highlights that the proposed channel pooling has the minimum overhead, while showing the overall best performance. As we discussed in Section 2.2, our channel pooling requires the least number of parameters for selective

| Feature subscription method | Parm. overhead (M) | Food | DTD | Action | Car | CUB |
|---|---|---|---|---|---|---|
| $1 \times 1$ Convolution | 14.0 | 82.82 | **74.73** | 78.29 | 77.07 | 73.20 |
| Channel-wise attention | 35.0 | **84.05** | 74.06 | **79.50** | 82.94 | **75.51** |
| Subscribing to all | 17.5 | 82.80 | 73.78 | 78.85 | 76.47 | 72.80 |
| **Channel pooling** | **5.5** | **84.06** | 72.37 | 78.48 | **83.79** | **75.81** |

Table 6: Parameter overhead and Top-1 accuracy (%) of **SNOW** for each feature subscription method.

feature subscription among other alternative methods, including $1 \times 1$ convolution, channel-wise attention (Woo et al., 2018), and subscribing to all features. Notably, the channel pooling approach shows the highest accuracy for most of the datasets (Food, Car, CUB), and still achieves comparable accuracy for Action dataset.

Although the $1 \times 1$ convolution uses more parameters than the channel pooling block, its accuracy is 6.72% less than channel pooling while slightly higher than the case of subscribing to all. This result indicates that the $1 \times 1$ convolution learns which channels are important for a new task, but its performance is worse than that of channel pooling in terms of all aspects, including accuracy, the number of required parameters, and computational overhead.

The channel-wise attention module (Woo et al., 2018) shows good accuracy for most of the datasets, however, its computation costs for inference are high. For the multi-task setup, its inferencing throughput is 1101.8 images/sec and the model parameter overhead is 35.0M while channel pooling achieves 4731.0 images/sec throughput with only 5.5M parameter overhead. Our channel pooling shows 4.3x higher throughput and 84.2% of parameter overhead reduction for multi-task inference than the channel-wise attention.

## 4 RELATED WORK

A large body of research has been done to address such challenges by exploring the various trade-off between training/serving/parameter efficiencies and accuracy. On the transfer learning side, the drawbacks in popular schemes such as feature extraction (Donahue et al., 2014) or fine-tuning (Dauphin et al., 2012; Cui et al., 2018) are addressed in various efforts. Mudrakarta et al. (2019) enhanced the parameter efficiency by inserting multiple patch layers into a frozen source model to capture the task-specific features at a cost of increased training and serving overhead. Knowledge flow (Liu et al., 2019) proposed a method that trains a learner from multiple experts by gradually distilling the layer-wise features from experts into the student at a cost of increased critical execution path for both training and inference as well. FixyNN (Whatmough et al., 2019) revisited the conventional feature-extraction based transfer learning approach from the computation perspective by re-designing and fine-tuning the last few top layers, which could degrade the model accuracy substantially unless enough number of top layers are optimized. Recently, Guo et al. (2019) improved the accuracy of transfer learning by adaptively selecting feature extraction blocks per sample at a cost of extra policy network and low training throughput.

On the lifelong learning side, catastrophic forgetting has been researched from various angles. Weight regularizations during re-training to retain the old task performance were studied in Lee et al. (2016); Weinshall & Cohen (2018). Li & Hoiem (2018) mitigated the catastrophic forgetting without old datasets by regularizing the final hidden activations from all the old tasks. Progressive Neural Net (Rusu et al., 2016) in reinforcement learning proposed to propagate the feature maps in a columnized set of models for different tasks, which would greatly increase the total model size, not to mention slower inference performance due to the complex per-layer inter-column adapters and expensive lateral connections (Guo et al., 2019; Chaudhry et al., 2019).

## 5 CONCLUSION

In this paper, we propose SNOW, an innovative way of transfer and lifelong learning by subscribing knowledge of a source model for a new task through channel pooling. SNOW architecture enables the delta model to subscribe to the intermediate features from each layer of the source model in a uni-directional manner. In this manner, we can effectively train a model for a new task by updating the parameters of the delta model only while keeping the source model untouched. SNOW offers

excellent training/serving speed by sharing the same features of the source model across all the delta models, yet offers competitive prediction accuracy with design a simple yet effective *Channel Pooling*. In our comprehensive experiments with multiple datasets, SNOW shows the best balance between the computing performance and prediction accuracy over all the datasets among the state-of-the-art transfer and lifelong learning methods during both training and serving process.

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

# A APPENDIX

## A.1 CHANNEL POOLING

We present the channel pooling block in Section 2.2 with Figure 3. To better understand, we provide the details of how the channel pooling block works in an operation level as follows.

---
**Algorithm 1** Channel Pooling Forward Function

---
$w$: channel pooling weight ($N \times 1$ vector)
$x$: feature map from a source model ($N$ channels)
$N \to M$ pooling
**procedure** CHANNELPOOLINGFORWARD($x$)
    **if** training **then**
        rand_v $= w + \mathcal{N}(0, \sigma^2)$
        _, idx $= topK$(rand_v, $M$)
        $selected\_weight = w$[idx]
    **else**
        $selected\_weight$, idx $= topK(w, M)$
    **end if**
    $x = x$[idx]
    $x = x \otimes selected\_weight$       //$\otimes$: channel-wise scalar multiplication
    $x = BatchNorm(x)$   //optional
    $x = ReLU(x)$      //optional
    return $x$
**end procedure**

---

## A.2 EXPERIMENT DETAILS OF TRAINING PERFORMANCE

Figure 4 shows training performance in terms of accuracy, throughput, and memory footprint for each dataset and training method. In the following tables, we report precise numbers of the graphs in Figure 4 as well as validation accuracy curves during training for all datasets.

| Training approach | Top-1 (top-5) accuracy | | | | |
|---|---|---|---|---|---|
| | Food | DTD | Action | Car | CUB |
| FO | 70.76 (90.55) | 70.81 (91.65) | 76.41 (95.58) | 51.96 (78.68) | 65.96 (89.90) |
| FE | 82.24 (95.84) | 71.45 (92.40) | 73.97 (93.64) | 74.05 (94.12) | 67.16 (91.27) |
| FT | 87.63 (97.32) | 61.56 (85.01) | 75.40 (93.02) | 90.26 (98.62) | 77.93 (93.37) |
| MP | 81.97 (95.68) | 72.41 (92.88) | 81.51 (95.44) | 81.42 (96.00) | 77.22 (94.24) |
| PN | 86.19 (96.97) | 74.32 (92.34) | 79.99 (94.41) | 86.26 (97.71) | 76.51 (93.63) |
| **SNOW-256** | 84.06 (96.47) | 72.37 (92.02) | 78.48 (95.04) | 83.79 (96.94) | 75.81 (94.05) |
| **SNOW-128** | 84.20 (96.52) | 73.98 (93.06) | 79.28 (95.25) | 83.69 (97.04) | 75.51 (93.87) |

Table 7: Test accuracy of each approaches with 5 datasets.

| Training approach | Throughput (images/sec) \| Memory usage (GB) | | | | |
|---|---|---|---|---|---|
| | Food | DTD | Action | Car | CUB |
| FO | 1076.08 \| 4.12 | 1089.83 \| 4.40 | 1106.31 \| 4.38 | 1085.67 \| 4.24 | 1102.50 \| 4.63 |
| FE | 1205.84 \| 4.05 | 1205.84 \| 4.31 | 1230.77 \| 4.02 | 1209.26 \| 4.17 | 1219.63 \| 4.31 |
| FT | 356.84 \| 13.74 | 357.94 \| 13.74 | 359.13 \| 13.83 | 358.74 \| 13.75 | 358.24 \| 14.66 |
| MP | 391.36 \| 14.57 | 393.60 \| 14.68 | 394.25 \| 14.69 | 388.66 \| 14.67 | 392.96 \| 13.72 |
| PN | 189.69 \| 15.29 | 189.09 \| 15.60 | 189.50 \| 15.29 | 185.01 \| 15.72 | 187.24 \| 15.56 |
| **SNOW-256** | 947.80 \| 4.40 | 959.16 \| 4.40 | 949.55 \| 4.26 | 965.31 \| 4.25 | 970.06 \| 4.27 |
| **SNOW-128** | 871.34 \| 2.92 | 873.12 \| 2.87 | 868.97 \| 2.80 | 885.20 \| 2.87 | 880.94 \| 2.94 |

Table 8: Computational performance for training on a single GPU

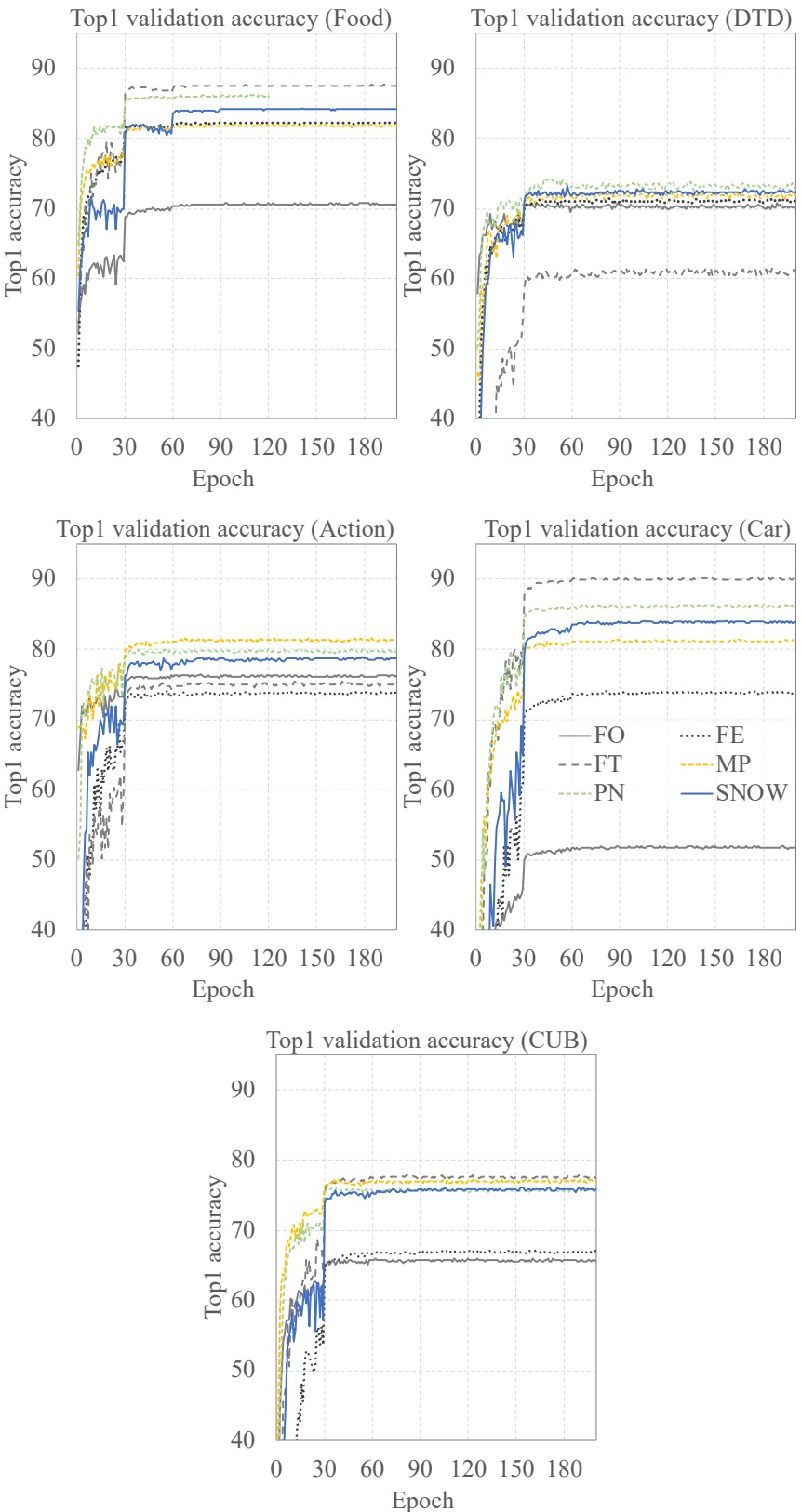

Figure 6: Validation accuracy curves during training for each dataset

### A.3 Learning without Forgetting

In our main experiment, we do not include the results from SNOW with Learning without Forgetting (**LF**) (Li & Hoiem, 2018) due to the catastrophic forgetting issue as shown in Table 9. Simply, the accuracies from LF for the experimented datasets are quite off from other competing algorithms, making it hard to put them on the same charts. We explored a few sequences to get the best outcome for LF and came up with the following in Table 9, which led to the accuracy changes for the datasets below. We used a batch size of 128, 0.005 learning rate, and $\lambda_o = 1$ as suggested in (Li & Hoiem, 2018). As you can see in the result for the training order A, once Action dataset is used, the accuracy against Car dataset drops significantly, as both are very heterogeneous (in terms of target domain and dataset size). Yet, adding CUB has somewhat limited impacts on Action. Also, we observe that the accuracy of LF depends on a training order. The accuracy from the training order A and training order B is significantly different before adding Food data. Overall, the combination of datasets in our experiment seems to be a very challenging scenario for LF in terms of catastrophic forgetting.

| Task | Training Order A (starting from ImageNet) | | | | |
|---|---|---|---|---|---|
| | $\rightarrow$ **Car** | $\rightarrow$ **Action** | $\rightarrow$ **CUB** | $\rightarrow$ **DTD** | $\rightarrow$ **Food** |
| **Car** | 81.87 | 28.40 | 22.78 | 22.44 | 15.31 |
| **Action** | - | 77.08 | 68.87 | 62.60 | 56.36 |
| **CUB** | - | - | 70.94 | 46.12 | 32.91 |
| **DTD** | - | - | - | 71.93 | 39.29 |
| **Food** | - | - | - | - | 79.71 |

| Task | Training Order B (starting from ImageNet) | | | | |
|---|---|---|---|---|---|
| | $\rightarrow$ **Action** | $\rightarrow$ **CUB** | $\rightarrow$ **DTD** | $\rightarrow$ **Car** | $\rightarrow$ **Food** |
| **Action** | 80.63 | 72.31 | 66.74 | 55.5 | 57.36 |
| **CUB** | - | 75.91 | 54.33 | 52.19 | 36.00 |
| **DTD** | - | - | 73.26 | 43.27 | 40.78 |
| **Car** | - | - | - | 73.77 | 16.33 |
| **Food** | - | - | - | - | 79.10 |

Table 9: Top-1 accuracy of Learning without Forgetting (**LF**)

**LF** overall showed the training throughput of 272.69 images/sec with 13.99GB memory footprint. The reason why **LF** is slower than **FT** is because **LF** needs to perform extra forward paths to compute the loss for the old tasks.

