# OpenReview forum: "SNOW: Subscribing to Knowledge via Channel Pooling for Transfer & Lifelong Learning of Convolutional Neural Networks"
_ICLR.cc/2020/Conference — Accept (Poster)_

### Official Review · AnonReviewer2 · 2019-10-21
**Official Blind Review #2**

**Rating:** 3

**Review:**

This paper attempts to tackle transfer learning and lifelong learning problem by subscribing to knowledge via channel pooling. The channel pooling is actually selecting the subsect of the feature map according to the way that prediction accuracy from the delta model can be maximized. Experiments show effectiveness of the proposed method.

Pros:
Overall, this paper is well written and easy to follow. The technique is sound and the problem studied in this paper is significant.

Cons:
1.	I do not think that the model proposed in this paper is able to tackle lifelong learning problem. The main reason is that lifelong learning basically requires only one model that will continue to learn from new tasks. After learning several new tasks, people hope this model can still perform well on the previous tasks as well as the current ones. However, in this paper, not only one model is learned. Instead, new models appear when new tasks are given, which does not meet the definition or requirement of lifelong learning. It only meets the requirement of transfer learning. The experimental results also validate my opinion since only one new task is given while in lifelong learning, continuous new tasks will come and the original model should perform well on all of them as well as on the old tasks.
2.	In Figure 4, the legend in the first picture will confuse the readers. I suggest the authors put it outside all the figures. Besides, the proposed method in the last picture is not the best. What do the authors want to convey by this picture?



**Experience Assessment:**

I have published one or two papers in this area.

**Review Assessment: Checking Correctness Of Derivations And Theory:**

I carefully checked the derivations and theory.

**Review Assessment: Checking Correctness Of Experiments:**

I carefully checked the experiments.

**Review Assessment: Thoroughness In Paper Reading:**

I read the paper thoroughly.

---

> ### Author Response · Authors · 2019-11-15
> **Response to Reviewer#2**
>
>
> C1. We appreciate your comments on SNOW regarding the number of models. One might assume that SNOW consists of multiple models as it has the source model and the “delta” models. But, the term “delta model” is used for an easier explanation about the expanded parts in the SNOW architecture. We argue that the entire architecture based on SNOW can be regarded as a single model (which just consists of various modules to deliver multi-task predictions) because a delta model alone is not semantically sufficient for an intended task when there already exists transferability from the source task to the target task. In fact, the delta and source models must efficiently cooperate as a single-engine to perform target tasks effectively (which is the key difference from a collection of models such as ensemble learning), which is the main contribution in this work.
>
> Expanding a single model as in SNOW is becoming a popular approach to address catastrophic forgetting in lifelong learning. For example, ProgressiveNet (Rusu, 2016) in our experiment can be considered as a single model even though the entire layer pipelines are duplicated for each new task as an expansion. Other various kinds of expansions have been published in major venues recently for lifelong learning, and we here provide a list of the latest representative publications with short summaries.
>
> -Lifelong Learning with Dynamically Expandable Networks [ICLR18]
> The authors considered lifelong learning simply as a special case of online or incremental learning, in the case of deep neural networks. The proposed approach in this paper is to partially expand the model capacity with new tasks.
>
> -Autonomous Deep Learning: Continual learning approach for dynamic environments [ICDM19]
> A fully elastic deep neural network (DNN), namely Autonomous Deep Learning (ADL) is proposed where the new hidden layers can be dynamically added under the lifelong learning paradigm.
>
> -Scalable Recollections for Continual Lifelong Learning [AAAI19]
> A small auto-encoder per new task is attached for the experience replay purpose for multi-task lifelong learning (instead of episodic memories).
>
> -Continual Palmprint Recognition Without Forgetting [ICIP19]
> This paper proposes to use reinforcement learning to dynamically expand the neural network when facing newly registered palmprints.
>
> -Lifelong Learning Starting from Zero [ICAGI19]
> This work proposes to add new nodes/neurons for expansion (which adds new nodes to memorize new input combinations) and generalization (which adds new nodes that generalize from existing ones).
>
> C2. Thanks for suggestions. We tried to place the legend outside the figures, but the space limitation makes it very hard. As an alternative solution to prevent any confusion on the readers, we overlay the legend over a gray bounding box which increases the readability as well. Hope this would work for you and future readers.
> The purpose of the last picture in Fig. 4 is to show that SNOW requires a similar number of epochs for convergence as well. In the end, what matters most in practice is the wall-clock runtime which is num_epochs X total_samples/throughput. In some cases, it is possible that a certain approach may have a better throughput but require more epochs for convergence, eventually netting a longer end2end training time. Here, with the pictures in Fig. 4, we showed that the total training time of SNOW will be superior to PN, FT, MP, because of the same epoch count for convergence (after 60 epochs) and higher throughput. Note that we simply let all the algorithms run for 200 epochs to ensure that nothing is left for any algorithm. We have made this point clear in the revision (the last paragraph in Section 3.1).

---

### Official Review · AnonReviewer3 · 2019-10-22
**Official Blind Review #3**

**Rating:** 8

**Review:**

*** Increased to Accept from Weak Accept after author rebuttal and changes to the paper ***

This paper proposes a method, SNOW, for improving the speed of training and inference for transfer and lifelong learning. SNOW starts with a pre-trained, frozen source model, and trains delta models for target tasks which, at each layer, concatenate a small number of task-specific features with the top-K most useful subset of features in the corresponding layer in the source model. As long as the target tasks are sufficiently related to the source task, it allows for small delta models and a small additional parameter overhead in the form of one weight per source model feature map. While there are (i) some issues with the presentation of results for training efficiency, (ii) some question marks over the sensitivity of the model to hyper-parameters, and (iii) several grammatical errors / typos in the manuscript, if these can be addressed I recommend the paper for acceptance because it seems to strike a superior balance of efficiency (regarding memory usage and inference speed) and accuracy when compared to a number of baselines, and to my knowledge it is a novel approach.

Detailed comments:
* Section 2.2 - How is sigma (the exploratory noise added for feature selection during training) chosen and how sensitive is the approach to its value? It seems like it was fine-tuned, given that a different sigma is chosen for the Action dataset (several orders of magnitude difference). In practice, tuning sigma could significantly increase training time.
* It seems like the performance of only one run was plotted per hyperparameter setting - it would be informative to see a mean and standard deviation especially since the approach seems like it could be unstable for the wrong hyperparameter settings.
* Related to the previous point, how much do the top-K feature selections change throughout training? One would have thought that this could cause instability during training for a high sigma. If sigma is too low, you could end up with suboptimal feature selection.
* Figure 4 graphs are a bit misleading because the throughput on the x-axis is reported per GPU and the larger models all need 2 or more GPUs. While this is mentioned in the main text, it is still optically deceptive and the results are GPU-dependent - presumably if the GPUs had a larger memory, the larger models would not seems as slow. I think it would be clearer to plot images/sec on the x-axis or to rerun the experiments just using a single GPU.
* It is stated that  “[d]etermining K […] has a critical impact on both size and target accuracy in the target models”, where K is the number of feature maps in the source model that the delta model subscribes to in each layer. How sensitive is the accuracy exactly? Can this be quantified or discussed in more detail?
* Furthermore, how sensitive is the performance to the number of target-model-specific features at each layer?
* Different learning rate schedules were used for SNOW and baselines - initial lr for SNOW is 1.0, while for all other models it is 0.1. Was it checked whether the baselines improve when they are run with an initial lr of 1.0? Was this hyperparameter more heavily tuned for SNOW than for the baselines?
* Since the source model is fixed, the applicability of the approach to lifelong learning is heavily dependent on the usefulness of the source model to subsequent tasks. If it is not, then one will have to incorporate large delta models. Furthermore, there can be no transfer between the tasks trained in the delta models.

Grammatical errors / suggestions:
* Page 1, first line: “hallmark” doesn’t make sense in this context - maybe “key objective” or “goal”?
* Page 1, 2nd paragraph, first line: “wee” -> “we”.
* Page 1, 2nd paragraph, line 6: “best top-K” -> either “K best” or “top K"
* Page 2, last paragraph, first line: “three folds” -> “threefold"
* Section 2.1, line 2: “pooing”->”pooling”. Same typo on Page 4, last line.
* Page 6, line 1: “training from the scratch” -> “training from scratch"
* Page 6, line 9: “more 6x than” -> “6x more than"
* Overall, the manuscript needs to be proofread a few times.

**Experience Assessment:**

I have read many papers in this area.

**Review Assessment: Checking Correctness Of Derivations And Theory:**

I assessed the sensibility of the derivations and theory.

**Review Assessment: Checking Correctness Of Experiments:**

I assessed the sensibility of the experiments.

**Review Assessment: Thoroughness In Paper Reading:**

I read the paper thoroughly.

---

> ### Author Response · Authors · 2019-11-15
> **Response to Reivewer#3 - Part 1**
>
>
> C1. Thanks for the comment. To demonstrate the effects of different sigma values, we did apply different values to the car dataset, and here is the result:
>
> sigma	1e-1        1e-3	        1e-5	1e-7
> ---------------------------------------------------------
> top1	81.27	83.79	83.45	83.35
> top5	95.85	96.94	96.99	96.88
>
> As you can see too large or too small sigma values can lead to sub-optimal predictive power. Therefore, it is important to develop a method to find good sigma values, as the reviewer pointed out. We're currently researching in that direction: one idea we have is to examine the early weight distribution (i.e., after a few epochs) and then determine the sigma value to balance out exploration and stabilization. We discussed this point in Section 3.2.
>
>
> C2. Thanks for the valuable suggestions. We have measured SNOW-256 (with batch size 256) accuracy again over 5 times on each dataset. We found that the avg top-1 accuracy is in fact slightly better than ones in the submission draft. Here is the accuracy distribution and we have updated the draft with the avg/std numbers accordingly.
>
>               Food             DTD         Action      Cars        CUB
> --------------------------------------------------------------------------
> avg.	84.06	    72.37	    78.48       83.79	    75.81
> std	        0.124          0.520       0.265       0.181       0.297
>
> Note that SNOW-128 (with batch size 128) results are also based on the 5 runs.
>
>
> C3. The top-K feature selections change very frequently at the beginning of the training and get stabilized with more epochs. Figure 5 in Section 3.2 shows which features are selected (in the solid vertical lines) during the training of CAR dataset under the same configuration/hyper-parameters in Section 3.1. The top X-axis represents the channel indices, and the left Y-axis represents the training progress (in terms of iteration). The horizontal dotted lines indicate the start of the next epoch. You can see that some channels join and leave the top-K frequently (i.e., small dots) yet some stay in the top-K consistently, getting more stable as training continues.
>
> Regarding the comment on sigma, please refer to the provided table in C1.
>
> C4. Thanks for the comments, and we agreed with the reviewer and updated the paper accordingly. We wanted to normalize the comparison over the typical mini-batch size for the tested datasets, which made PyTorch split the training over multiple GPUs. When two GPUs are used for training, the communication between GPUs can incur some overheads. However, our platform has GPUDirect over NVLink2 between GPUs which has 160GB/s bandwidth, thus the impact should have been rather limited.
>
> Per the reviewer's suggestion, we refreshed all the experiments under the single GPU constraint. In detail, we reduced the batch sizes of MP, FT, PN until it fits into one GPU. Additionally, we tested SNOW with a smaller batch size (128 from 256) to ensure that SNOW still offers advantages with that configuration. As a result,  for the example of the Car dataset, the throughput gap between SNOW and PN decreases from 6x to 5.2x. Figure 4 and Table 8 (in the Appendix) are all accordingly updated.
>
>
> C5.  Thank you for the input. As suggested, we experimented with the CAR dataset to study the effect of different Ks. Our current finding shows that the accuracy is somewhat sensitive to the K values, and it seems there could be some sweet spot for K: we believe oversubscription may introduce unwanted noises to the delta model (in addition to the size/compute overhead). We elaborated more in Section 2.2 and added results to Section 3.2.
>
>    K                  |    N/4         N/8         N/16
> ---------------------------------------------------
>   accuracy      |    83.10       83.79       83.39
>
>
> C6.  We again used the CAR dataset to study the performance sensitivity to the number of target-model-specific features and showed the results below.
>
> target-specific  |
> feature count   |    M/4         M/8         M/16
> ---------------------------------------------------
>   accuracy          |    79.02       83.79       80.36
>
> It shows that the performance is sensitive to the delta model size, and clearly exposes the existence of ideal size. It is obvious that the same rule of thumb in neural network architecture design applies here too: having too few target-specific features hurt accuracy, but having too many (or too big delta net) does hurt as well because the number of target-specific samples may be relatively too small. We added discussion and result in Section 3.2.

---

> > ### Author Response · Authors · 2019-11-15
> > **Response to Reviewer#3 - Part 2**
> >
> >
> > C7.  Thank you for suggesting another important comparison to re-validate our results. We used it 0.1 as it is a standard learning rate for resnet50 family in many publications. The reason why we picked a larger learning rate for SNOW was to tune our weights fast enough to stabilize the training earlier. We have not really tuned the hyper-parameters for SNOW (i.e., learning rate/schedule) yet, which is one of our future to-do items. To ensure no mistake, we applied lr=1.0 with batch size 256 to all baselines on the CAR dataset, and the results on Top-1 accuracy are here.
> >
> >  	          |   FO	          FE	        FT	   MP	   PN
> > ------------------------------------------------------------------------
> > accuracy  |	51.95	 49.12    1.24	  4.79	 59.02
> >
> > It confirms that lr=0.1 was a much superior choice for other algorithms to lr=1.0.
> >
> > C8. Your statement is correct that a larger delta model would be needed for such cases, as the delta model needs to generate more features by itself. We highlighted your point in the first paragraph of Section 2.1.
> >
> > C9. Grammatical errors / suggestions: All corrected, thank you.

---

> > > ### Comment · AnonReviewer3 · 2019-11-15
> > > **Good improvements to the paper - increasing to Accept**
> > >
> > > Thank you for your detailed reply and the changes to the paper. The hyperparameter sensitivity and the robustness of the model are now clearer, and I am happy to increase my score to Accept.

---

### Official Review · AnonReviewer1 · 2019-11-02
**Official Blind Review #1**

**Rating:** 8

**Review:**

After rebuttal:
Authors have addressed all my doubts. I recommend accepting this paper.

=============================
Before rebuttal:
Summary:

This paper proposes a new way to do transfer learning. Specifically, authors first train a big source ConvNet and then for each task, they train a small ConvNet in which each layer subscribes to some k channels in the corresponding layer of the source ConvNet. Authors show that this model works better than methods that fine-tune the last few layers of the source network and performs close to costlier methods like progressive networks but with lesser parameters and higher throughput. Experiments on 5 tasks verify their claim.


My comments:

Overall, this is a very interesting paper.

1. This is an interesting model to do transfer or lifelong learning but only for ConvNet architectures with image data. To avoid overstating the results, I request the authors to highlight this limitation in both the title and the abstract.
2. Page 3, para starting with “In detail”: Is the ResNet50 for delta model pre-trained or not? I know it is not pre-trained based on future paragraphs. But it is good to clarify it here.
3. Sharing the same source network across multiple tasks during inference time is useful only when all the tasks take the same input. This is a very restricted application. This needs to be elaborated and highlighted in the paper.
4. I would like to see the LF results included in the paper even though it has catastrophic forgetting issues.
5. In Figure 4, the x-axis represents training throughput or inference throughput? I guess it is training throughput. Also, are the models trained for all the tasks in parallel (as described in serving all the tasks at once section) or separately? Even though I can guess answers for these, it is better to make these explicit in the paper for the benefit of the readers.
6. It is never a good idea to show test curves for a task. Please remove the test curves from Figure 4. Instead, use a separate validation set and show validation curves.
7. Are the authors willing to release the code to reproduce their results?

Minor comments:

1. Section 1, second para, 1st line: “wee” should be “we”
2. Table 1: Fix grammar in MP description.


**Experience Assessment:**

I have published one or two papers in this area.

**Review Assessment: Checking Correctness Of Derivations And Theory:**

N/A

**Review Assessment: Checking Correctness Of Experiments:**

I carefully checked the experiments.

**Review Assessment: Thoroughness In Paper Reading:**

I read the paper thoroughly.

---

> ### Author Response · Authors · 2019-11-15
> **Response to Reviewer#1 (updated with more LF data)**
>
>
> C1. Thanks for the comment. We reflected your request in both the title and abstract of the revision.
>
> C2. Thanks for helping us clarify an important aspect of our scheme. We clearly indicated that the delta model is to be trained in the paragraph you mentioned as well as the paragraph before it ("individual to-be-trained delta model) to avoid any confusion.
>
> C3. We appreciate your suggestion. We have elaborated the reviewer's point in the paper by adding a new paragraph in Section 2.1.
>
> C4.  Thanks for the comments. We checked out our implementation to ensure the correctness and performed hyper-parameter tuning to get the best performance for LF. We believe that our test case is extremely challenging for LF, because our five datasets do have very different distributions. Note that the multi-task scenario in the original LF paper, in fact, uses multiple subsets of a single VOC dataset.
>
> Our training is still running and we will append the table as soon as the training job is over (in a few hours). We will include the results to the appendix of the revision at the same time.
> ====> We have completed the runs except for Food (which needs another 10+ hours). We will add the entire outcome to the final version ASAP.
>
> We explored a few sequences to get the best outcome for LF,  came up with the following,  which led to the accuracy changes for the datasets below.
>
>    sequence ImgNet    ->Car     ->Action   ->CUB    ->DTD
>                      -----------------------------------------------------------
>                         Car      |  81.87     28.40      22.78     22.44 .  #Car Top1 accuracy drops as new tasks are being added.
>    accuracy    Action  |    X          77.08      68.87     62.60
>                        CUB      |   X            X            70.94     46.23
>                        DTD      |   X            X              X          71.93
>                     -------------------------------------------------------------
>
> As you can see, once Action dataset is used, the accuracy against Car dataset drops significantly, as both are very heterogeneous. Yet, adding CUB has somewhat limited impacts on Action. Overall, the combination of datasets in our experiment seems to be a very challenging scenario for LF in terms of catastrophic forgetting. We agree with the reviewer's suggestion to keep the LF result in the paper, as perhaps this can serve as a good example to study for fellow researchers.
>
> Regarding the computational performance, LF overall showed the training throughput of 272.69 images/sec with 13.99 GB memory footprint. The reason why LF is slower than FT is LF needs to perform extra forward paths to compute the loss for the old tasks.
>
> C5. Thank you for the question and comments. The x-axis in Figure 4 represents training throughput. We double-checked the caption of Figure 4 to ensure that it states training. The models are trained separately in a sequential manner, and we explicitly stated it at the beginning of Section 3.1.
>
> C6. Thanks for pointing out this. In fact, this was our typo, and the graph is indeed the validation curve. We have fixed it in the revision.
>
> C7. We are in the process of obtaining the clearance for code-release. In the meantime, the pseudo-code in Appendix A.1 should be sufficient enough for anyone to try out our channel pooling idea (i.e., it is already almost a python code snippet).
>
> C8. Minor comments: all corrected, thank you.

---

### Author Response · Authors · 2019-11-15
**Change overview**

Dear reviewers,
To address your concerns and comments, we have revised the draft with the following major changes. In the uploaded version, we have addressed all the cosmetic errors/typos.
1. We performed a new set of experiments with the 1-GPU constraints and refreshed Fig 4.
2. We introduced a new Section 3.2 to discuss the hyper-parameter sensitivity.
3. We captured the Top-K changes during training in Figure 5.

Thank you.
Authors.

---

### Decision · Program_Chairs · 2019-12-19

**Decision:**

Accept (Poster)

**Comment:**

This paper proposes a method, SNOW, for improving the speed of training and inference for transfer and lifelong learning by subscribing the target delta model to the knowledge of source pretrained model via channel pooling.

Reviewers and AC agree that this paper is well written, with simple but sound technique towards an important problem and with promising empirical performance. The main critique is that the approach can only tackle transfer learning while failing in the lifelong setting. Authors provided convincing feedbacks on this key point. Details requested by the reviewers were all well addressed in the revision.

Hence I recommend acceptance.